# A Report Card on Prevention Efforts of COVID-19 Deaths in US

**DOI:** 10.3390/healthcare9091175

**Published:** 2021-09-07

**Authors:** Ramalingam Shanmugam, Lawrence Fulton, Zo Ramamonjiarivelo, José Betancourt, Brad Beauvais, Clemens Scott Kruse, Matthew S. Brooks

**Affiliations:** School of Health Administration, Texas State University, San Marcos, TX 78666, USA; Lf25@txstate.edu (L.F.); zhr3@txstate.edu (Z.R.); jose.a.betancourt@txstate.edu (J.B.); bmb230@txstate.edu (B.B.); Scottkruse@txstate.edu (C.S.K.); Mbrooks@txstate.edu (M.S.B.)

**Keywords:** modeling, data analysis, assessment, effectiveness, incidence rate, restriction

## Abstract

COVID-19 (otherwise known as coronavirus disease 2019) is a life-threatening pandemic that has been combatted in various ways by the government, public health officials, and health care providers. These interventions have been met with varying levels of success. Ultimately, we question if the preventive efforts have reduced COVID-19 deaths in the United States. To address this question, we analyze data pertaining to COVID-19 deaths drawn from the Centers for Disease Control and Prevention (CDC). For this purpose, we employ *incidence rate restricted Poisson (IRRP)* as an underlying analysis methodology and evaluate all preventive efforts utilized to attempt to reduce COVID-19 deaths. Interpretations of analytic results and graphical visualizations are used to emphasize our various findings. Much needed modifications of the public health policies with respect to dealing with any future pandemics are compiled, critically assessed, and discussed.

## 1. Introduction

The worldwide spread of the novel severe acute respiratory syndrome coronavirus 2 (SARS-Cov-2), which causes the infectious coronavirus disease 2019 (COVID-19), has resulted in millions of deaths around the world since the detection of the first COVID-19 case in Wuhan, China in December 2019 [1,2,3,4]. As of 22 July 2021, the Center for Systems Science and Engineering [2] at Johns Hopkins University reported over 192 million COVID-19 confirmed cases and over 4.1 million deaths worldwide [2]. The United States has been greatly affected by this pandemic with over 34.2 million infected individuals and over 610,000 deaths [2]. Nationwide, a number of recommended efforts have been implemented to contain the spread of the virus, such as frequent and thorough handwashing, no face-touching, wearing of masks, home confinement, social distancing, local attitudes of individuals, routine behaviors of those visiting bars to drink as a social interaction, staying at home in times of delta variants, bans on social gatherings, thorough disinfection of households and other facilities, testing, social distancing, contact tracing, and quarantining those exposed to the virus.

This paper aims to assess how the state-level deaths due to COVID-19 correlate with medical interventions and social distancing as restrictions by the states in the US, and how important and timely are they. For this purpose, we select an appropriate underlying model for the data. The selected model is the *incidence rate restricted Poisson (IRRP)* because the death incidences are Poisson type but with constraints on them due to medical interventions and social distancing factors. The strengths of these restrictive factors, in a collective sense, can be associated with a reduction of the possible death rate range. Otherwise (that is when the factors are weaker or non-existent), the possible domain death rate expands, according to our selected model. More details follow after Equation (1) in the article. To use simpler regression techniques, one needs pertinent data on potential covariates/predictors related to COVID-19. Unfortunately, the COVID-19 database does not have information on covariates/predictors. Data for the United States about the number of people who practice frequent and thorough handwashing, avoid face-touching, wear masks, comply with home confinement, respect social distancing, as well as the data on individuals who do not comply with Center for Disease Control (CDC) guidelines, especially for those who contracted the virus are not available. The data collection is costly, time-consuming, and in fact not advisable during the pandemic. Searching and selecting an appropriate model, as done in this study is the only viable alternative to achieve our aim. The components of the selected model are explained after Equation (1), later in the article. Furthermore, the death rate due to COVID-19, the restriction level due to medical intervention, and the social or preventive constraints are inter-relative with co-movement in a chain relationship over the months, the Bayesian concept and its tools are quite fitting to extract and interpret evidence from the data to capture the health dynamics in the United States. The prior conjugate plays a vital role in the application of the Bayesian approach, as this is explained with reasons and details later in the article centering on the methodology section.

Data have shown some variations in SARS-Cov-2 infection and death-related rates across all affected 188 countries [2]. With respect to the United States, data have indicated some variations across states as well as across counties within each state. The purpose of this study is to assess the factors associated with virus containment (lower infection rate) and lower COVID-19 death rate among counties in the United States. A recent study concluded that “at a state level, in the US, the “Stay in Place” (SIP) order is effective at decreasing the compound growth rate of COVID-19. The U.S. counties that have the largest impact from a SIP order are ones with a large population or a high population/density” [5]. This study intends to compare the efforts taken by neighboring counties and assess if those efforts significantly reduce the SARS-Cov-2 infection rate and COVID-19 death rate [5]. There is a need for a capability to aggregate state-level COVID-19 data into metropolitan areas and display these data in an interactive dashboard that updates in real-time. The purpose of this website https://coronavirus.jhu.edu/map.html (accessed on 1 July 2021) should be to make this information more accessible to the public and to allow for a more granular assessment of infection spread and impact [6].

## 2. Background Knowledge of COVID-19

SARS-Cov-2 spread was believed to have originated from animal-to-human transmission via bats, and then from human-to-human [7]. The virus spreads from human-to-human via the saliva, nose secretion, or breath of the contaminated person. When the contaminated person talks, sings, coughs, or sneezes, the virus is expelled, spreads in the air, and enters the mouth, eyes, or nose of other people, and/or land on objects and surfaces. Also, the hands of the contaminated person may spread the virus if that person touches some objects. Both symptomatic and asymptomatic individuals who are infected can spread the virus [8]. COVID-19 symptoms include fever, dry cough, fatigue, nasal congestion, headache, conjunctivitis, sore throat, diarrhea, loss of taste or smell, skin rash, and fingers or toes discoloration. About 80% of infected individuals have mild symptoms and recover without hospitalization. Those who become seriously ill develop respiratory distress and require hospitalization [9].

Factors such as poverty, unemployment (socioeconomic status domain), crowded housing, and vehicle access (housing and transportation domain) were associated with increased COVID-19 diagnoses and deaths in urban areas [10,11]. About 33% of rural counties are highly susceptible to COVID-19, driven by older and health-compromised populations, and scarcity of health care facilities for the elderly. More precisely, the major vulnerabilities of rural counties are associated with a physician shortage, a higher proportion of individuals with a disability, and a larger uninsured population, compared with urban counties. Although lack of a data network limits broadband services, the existence of a cellular network still enables mobile health (mHealth, part of telemedicine) interventions for preventive and other health services that exchange health information for improving health outcomes through simple message service (SMS) [12,13]. Health enables communication activities such as data storage, retrieval, and communication pertaining to patients’ healthy behaviors, choices, and medical diagnoses [12]. In addition, lack of social capital and social services may also hinder recovery from the pandemic [13]. Data also indicated some health disparities with regards to COVID-19 diagnoses and deaths. For instance, nearly 20% of U.S. counties are disproportionately black, and yet, they accounted for 52% of COVID-19 diagnoses and 58% of COVID-19 deaths nationally. State-level comparisons can both inform COVID-19 responses and identify epidemic hot spots. Social conditions, structural racism, and other factors may elevate the risk for COVID-19 diagnoses and deaths in black communities [14].

Ongoing challenges to protect youth and family well-being and address workforce needs following the pandemic include (a) addressing digital disparities and lack of access to computer technology among families and providers; (b) addressing the need for trauma-informed care and mental health services for youth and families; (c) providing ongoing specialized well-being services to frontline providers and their families; (d) addressing the structural determinants of health among highly vulnerable families through employment and eviction protections, housing provisions, healthcare access, and strengthening the safety net; and (e) ensuring that highly vulnerable youth do not fall through the cracks given disruptions to services that many children in state systems disproportionately rely upon. Despite these challenges, opportunities exist to harness the coordinated efforts of state departments, agencies, community providers, and academic centers to narrow—instead of widening—the equity gap so that children and families can emerge from this pandemic stronger [15].

## 3. Methods

### 3.1. Data & Sample

Data for analysis and discussion in this article were collected from: https://data.cdc.gov/NCHS/Provisional-COVID-19-Death-Counts-in-the-United-St/kn79-hsxy website (accessed on 1 July 2021). There is a total of 3195 counties in the United States. Among the variables that exist in the database, we consider the state, the monthly number of deaths from COVID-19 from March 2020 till June 2021. We computed the average, $\bar{y}$ and variance, $s_{y}^{2}$ for of the number deaths due to COVID-19 in the state. The large variance is indicative of the existence of *heterogeneity* of the pandemic in the state, we noticed that in every state, the variance is higher than the mean as it is a requirement for the selected *incidence rate restricted Poisson (IRRP)* as an underlying model for data analysis.

### 3.2. Efforts

#### 3.2.1. Probability Model Justification

Let *Y* be an uncertain number of deaths due to COVID-19 in a month in a state with its deaths rate $0<\theta <\frac{1}{\gamma }$, in which the restriction parameter γ≥0 portrays the collective impact of various preventive efforts on the death rate θ. Preventive efforts consist of social distancing, face masking, hand-washing, and total lock-down, among others. In other words, the *odds* (that is, the ratio of the chance of having one or more deaths over the chance of no death) due to COVID-19 in a state is $\theta e^{-\gamma \theta }$, which is smaller under the existence (that is, γ>0) of preventive efforts than otherwise (that is, γ=0). The research goal of this article is to extract the data evidence to statistically check whether γ=0 or γ>0. Naturally, an underlying probability model [16] (connecting the death rate, θ and the restriction level, γ) for the chance-oriented mechanism of the COVID-19 pattern in a state is a necessity. For this purpose, we consider the *incidence rate restricted Poisson (IRRP) distribution* in Equation (1). That is,
(1)$$\mathrm{Pr}(Y|\theta ,\gamma )={\left(\theta e^{-\gamma \theta }\right)}^{y-1}{\left(1+\gamma y\right)}^{y-1}e^{-\theta }/y!;y=0,1,2,\ldots ;0<\theta <\frac{1}{\gamma }$$

The expected number of deaths due to COVID-19 from (1) is $E(Y|\theta ,\gamma )=\frac{\theta }{\left(1-\gamma \theta \right)}$ (that is, the expected deaths is a ratio of death rate over a function of both the death rate and restriction) and the variance is $Var(Y|\theta ,\gamma )=\frac{1}{{\left(1-\gamma \theta \right)}^{2}}E(Y|\theta ,\gamma )$ (that is, the fluctuation in the number of deaths is proportionally inflated by the square of a function of both the death rate and the restriction level). The relationship between the variance and the expected number of deaths suggests that the variance is larger than the expected deaths (because γθ<1) in a COVID-19 scenario. Until then, why not model and learn from the state data in which the variance is larger than the expected number of deaths? Data indicated that the variance is larger than the mean in every state. It is worth pointing out in this context that the variance is a measure of heterogeneity. The number of COVID-19 deaths is volatile in every state, and it is interesting to realize that no state in the US is safe with respect to this pandemic.

There are two alternative ways to estimate the death parameter θ and the restriction parameter γ from the data. One method is the moment estimator, and it yields ${\hat{\theta }}_{moment}=\frac{{\bar{y}}^{3/2}}{s_{y}}$ and ${\hat{\gamma }}_{moment}=\frac{\left(1-\sqrt{\frac{\bar{y}}{s_{y}^{2}}}\right)}{\hat{\theta }}$. A virtuous (because of its invariance property) method is the maximum likelihood estimator (MLE) and it is the simultaneous solution of $\sum_{i=1}^{n}\frac{y_{i}(y_{i}-1)}{\left|{\hat{\theta }}_{mle}(y_{i}-\bar{y})\right|+y_{i}\bar{y}}=n$ and ${\hat{\gamma }}_{mle}=\left|\frac{1}{{\hat{\theta }}_{mle}}-\frac{1}{\bar{y}}\right|$. The invariance property indicates that an MLE of a function of the parameters is simply the function of the MLE for the parameters. In the COVID-19 data (due to their largeness), the MLEs and the moment estimators are almost the same.

Of main interest is whether the estimate, $\hat{\gamma }$ is statistically significant. If so, it is an attestation that all preventive efforts had been effective in placing a cap on the COVID-19′s death rate. Otherwise, there is a need to refine the existing and/or introduce better new preventive efforts to control COVID-19′s death rate according to the data. To perform this, we resort to a hypothesis testing procedure as devised [16]. The *p*-value of rejecting the null hypothesis $H_{o}:\gamma =0$ is
(2)$$\mathrm{Pr}[-2\{\hat{\theta }\hat{\gamma }\bar{y}+n(\hat{\theta }-\bar{y})-n\bar{y}\ln(\frac{\hat{\theta }}{\bar{y}})\}-\hat{\gamma }\{(n-1)s_{y}^{2}+n\bar{y}(\bar{y}-1)\}>\chi_{1df,pvalue}^{2}]=p-value$$

The (statistical) power of accepting a specific alternative hypothesis $H_{1}:\gamma =\gamma_{1}$ is
(3)$$Power\approx \mathrm{Pr}[\chi_{\delta_{\gamma_{1}}df}^{2}>\frac{\chi_{1df,\alpha }^{2}}{\left\{1+\frac{\delta_{\gamma_{1}}}{1+\delta_{\gamma_{1}}}\right\}}]$$
where α is the significance level at which the null hypothesis is rejected and $\delta_{\gamma_{1}}=2\gamma_{1}^{2}(\frac{n{\hat{\theta }}_{1}^{2}}{1+2\gamma_{1}})$ is the non-centrality parameter.

#### 3.2.2. Bayesian Fabric and Second Layer of Data Analysis for Each State

Bayesian analysis is catching the attention of epidemiologists, public health researchers, and policymakers (O’Hagan [16]). It is more prevalent in COVID-19 discussions because the death rate and its restriction level are y changing on a daily basis. That is exactly a reason to utilize a Bayesian approach. In our scenario, the death rate, θ, is fluctuating daily, qualifying to hold a probability pattern. The Bayesian concept involves the likelihood $L(y_{1},y_{2},\ldots ,y_{n}|\theta ,\gamma )$ for collecting data $y_{1},y_{2},\ldots ,y_{n}$ at a specified death rate in a day at a location, a compatible prior probability density function, π(θ) of the randomly fluctuating daily death rate, and a posterior probability density function, $\pi (\theta |y_{1},y_{2},\ldots ,y_{n})$ as updated information on the death rate in the aftermath of the collected data. We arbitrarily select a compatible prior $\pi (\theta )=(\frac{\phi }{\theta }){\left(\gamma \theta \right)}^{\phi };0<\theta <\frac{1}{\gamma };\gamma \geq 0$, where ϕ>0 is pronounced as a hyperparameter in the Bayesian approach. It is clear that the prior π(θ) is a bona fide probability density function (pdf) because π(θ)>0 and $\int_{0}^{1/\gamma }(\frac{\phi }{\theta }){\left(\gamma \theta \right)}^{\phi }d\theta =1$. For example, when the restriction level parameter, γ=1, the prior pdf π(θ) looks like in Figure 1.

In a Bayesian approach, even if the prior pdf is selected to non-match, the likelihood does more often moderate any incorrect prior estimate at the time of posteriority. The prior expected value and prior variance are respectively $E(\theta )=\frac{\phi }{\gamma (\phi +1)}$ and $Var(\theta )=\{\frac{1}{\gamma (1+\phi )(2+\phi )}\}E(\theta )$, where the balancing factor $\left\{\frac{1}{\gamma (1+\phi )(2+\phi )}\right\}$ illustrates the relationship between the variance and the mean. Their relationship is sketched in Figure 2. The posterior pdf is proportional to
$$\pi (\theta |y_{1},y_{2},\ldots ,y_{n};\gamma ,\phi )=N(y_{1},y_{2},\ldots ,y_{n};\phi ,\gamma )\pi (\theta )L(y_{1},y_{2},\ldots ,y_{n}|\theta ,\gamma )$$
where the normalizer $N(y_{1},y_{2},\ldots ,y_{n};\phi ,\gamma )$ is chosen to make the function $\pi (\theta |y_{1},y_{2},\ldots ,y_{n};\gamma )$ of the death rate is a bonafide posterior pdf function. That is,
(4)$$N(y_{1},y_{2},\ldots ,y_{n};\phi ,\gamma )=\frac{{\left\{n(1+\gamma \bar{y})\right\}}^{n\bar{y}+\phi }}{\Gamma (n\bar{y}+\phi )\Phi_{\chi_{(n\bar{y}+\phi )df}^{2}}(2n\{\frac{1}{\gamma }+\bar{y}\})}$$
where $\Phi_{\chi_{(n\bar{y}+\phi )df}^{2}}(2n\{\frac{1}{\gamma }+\bar{y}\})$ is the cumulative chi-squared distribution function up to the argument $2n\{\frac{1}{\gamma }+\bar{y}\}$ with degrees of freedom (df)$\left(n\bar{y}+\phi \right)$. Hence, the posterior pdf is therefore
$$\pi (\theta |y_{1},y_{2},\ldots ,y_{n};\gamma ,\phi )=\frac{{\left\{n(1+\gamma \bar{y})\right\}}^{n\bar{y}+\phi }}{\Gamma (n\bar{y}+\phi )\Phi_{\chi_{(n\bar{y}+\phi )df}^{2}}(2n\{\frac{1}{\gamma }+\bar{y}\})}e^{-n(1+\gamma \bar{y})\theta }\theta^{(n\bar{y}+\phi )-1};0<\theta <\frac{1}{\gamma }$$

The posterior expected death rate is
(5)$$E(\theta |y_{1},y_{2},\ldots ,y_{n};\gamma ,\phi )=\{\frac{n\bar{y}+\phi }{n(1+\gamma \bar{y})}\}\{\frac{\Phi_{\chi_{(n\bar{y}+\phi +1)df}^{2}}(2n\{\frac{1}{\gamma }+\bar{y}\})}{\Phi_{\chi_{(n\bar{y}+\phi )df}^{2}}(2n\{\frac{1}{\gamma }+\bar{y}\})}\}\approx \{\frac{n\bar{y}+\phi }{n(1+\gamma \bar{y})}\}$$
and posterior variance
(6)$$\begin{array}{l} Var(\theta |y_{1},y_{2},\ldots ,y_{n};\gamma ,\phi )=\{\frac{n\bar{y}+\phi }{n(1+\gamma \bar{y})}\}\{\frac{n\bar{y}+\phi +1}{n(1+\gamma \bar{y})}\}\frac{\Phi_{\chi_{(n\bar{y}+\phi +2)df}^{2}}(2n\{\frac{1}{\gamma }+\bar{y}\})}{\Phi_{\chi_{(n\bar{y}+\phi )df}^{2}}(2n\{\frac{1}{\gamma }+\bar{y}\})}\}\\ -\{\frac{n\bar{y}+\phi }{n(1+\gamma \bar{y})}{\}}^{2}\{\frac{\Phi_{\chi_{(n\bar{y}+\phi +1)df}^{2}}(2n\{\frac{1}{\gamma }+\bar{y}\})}{\Phi_{\chi_{(n\bar{y}+\phi )df}^{2}}(2n\{\frac{1}{\gamma }+\bar{y}\})}{\}}^{2}\\ \approx \{\frac{1}{n(1+\gamma \bar{y})}\}E(\theta |y_{1},y_{2},\ldots ,y_{n};\gamma ,\phi ) \end{array}$$

We define the vulnerability index of the COVID-19 in a month at a state as a ratio
(7)$$Vu\ln erability=\frac{Var(\theta |y_{1},y_{2},\ldots ,y_{n};\gamma ,\phi )}{Var(\theta |y_{1},y_{2},\ldots ,y_{n};\gamma ,\phi )+E(\theta |y_{1},y_{2},\ldots ,y_{n};\gamma ,\phi )+2{\left\{E(\theta |y_{1},y_{2},\ldots ,y_{n};\gamma ,\phi )\right\}}^{2}}$$

This vulnerability (7) is computed for each month since March 2020 till June 2021 and compared with each other below.

### 3.3. Learning and Warning from COVID-19 Data Evidence

We could learn from the data analytic results and graphical visuals that are displayed below. Figure 3 (the displayed and not displayed alphabetical states) provides an overall impression of the COVID-19 death pattern since March 2020. Recall that there was no reported COVID-19 death in January and February 2020. However, in March 2020, there was a spike in COVID-19 deaths in California, New Jersey, Texas, New York City, and New York. Table 1 displays the estimate of the death rate $\hat{\theta }$, restriction rate $\hat{\gamma }$, hyper parameter $\hat{\phi }$, balance factor, the number of days n, average # deaths $\bar{y}$, posterior mean, $E(\theta |y_{1},y_{2},\ldots ,y_{n};\gamma ,\phi )$, posterior variance, $Var(\theta |y_{1},y_{2},\ldots ,y_{n};\gamma ,\phi )$ and vulnerability in Equation (7) to COVID-19 deaths. The *p*-values of the data-based estimate $\hat{\gamma }$ are less than 0.0001 meaning that the imposed restriction levels since March 2020 had been significant to reduce the COVID-19 death rate. The statistical power values to accept the hypothesis $H_{1}:\gamma =\hat{\gamma }$ are more than 0.99 meaning that the methodology works well for judging COVID-19 deaths since March 2020. Figure 4 and Figure 5 indicate that the months June 2020 through June 2021 formulate Cluster 1 and March, April, May 2020 formulate Cluster 2 with 81% of total variation explained by the two principal components.

Figure 6, Figure 7, Figure 8, Figure 9, Figure 10, Figure 11, Figure 12 and Figure 13 point out that the death rates $\hat{\theta }$, restriction levels $\hat{\gamma }$, hyperparameter $\hat{\phi }$, balancing factor, average number of deaths $\bar{y}$, posterior mean, variance, and vulnerability in US are progressively upward since March 2020 June 2021.

### 3.4. Limitations, Criticism, and Recommendations of Our Finding

All the above interpretations are as good as the authenticity of the COVID-19 data. There is a suspicion among the public that some of the deaths due to other confounding causes (recognized as co-morbidity in the medical literature) with COVID virus are incorrectly placed under the column for COVID-19 deaths. This suspicion is not verifiable for obvious reasons of impracticality. Hence, what we learned above is subject to limitations of data collection and authenticity The above learning is not wrong either. Some level of learning is more helpful than total ignorance of the COVID-19 pandemics.

## 4. Conclusions

Using appropriate data and a correct methodology to analyze and extract the evidence about what has gone correctly and what could have been done differently is the essence of this article. As we witnessed in the contents of this article, there had been an increasing death rate due to the COVID-19 pandemic and equally compatible stronger impact of medical and social interventions on the death rate. The performances in some states are similar while others differed significantly. However, the deaths in March, April, May 2020 clustered together, while all other months deviated into another cluster. This pattern becomes visible due to our data analysis. The death rates and the impact of medical and social interventions had consistently increased together over the months. Such a co-movement complicates our comprehension. There ought to have been a chain adjustment in their relationship and it was captured by the balancing factor. The chain relationship is the underlying reason for considering the aptness of the Bayesian concept and tools as they are used in the article. Interestingly, the balancing factor itself, as expected, consistently increased and it attested to the fact that the health system had been trying to control the pandemics effectively. In spite of the efforts to contain the pandemic, the vulnerability to death due to COVID-19 has been increasing over the months in a volatile manner because of the heterogenous nature of the US states.

A natural question to pose is that whether a pandemic like COVID-19 might recur. A reply probably depends on the philosophical orientation of the respondent. When the respondent feels that COVID-19 during years 2020–2021 is an anomaly, the reply might lead to a statement that no pandemic is likely in the future. On the contrary, when a respondent is a conspirator feeling that someone intentionally or by mistake leaked out the deadly virus from an experimental biologic research lab, the reply is likely to warn that a pandemic like COVID-19 would recur. However, every pandemic would alter the human lifestyle. Different types of vaccination would emerge. Virologists, pathologists, and public health professionals are going to be too busy. Preventive medicine would receive priority in public health. The general public might become too uncomfortable with free movements in society. Medical professionals could overutilize the resources to prepare vaccinations which could expire in time as they are not used. A disparity in the community might widen as the affordability to being healthy fluctuates more in favor of the rich than the middle or poor classes. A constant feeling of undergoing fear sabotages mental health and comfort living. The natural way of living takes up a backseat in comparison to subjecting to technology-oriented priority living.

## Figures and Tables

**Figure 1 healthcare-09-01175-f001:**
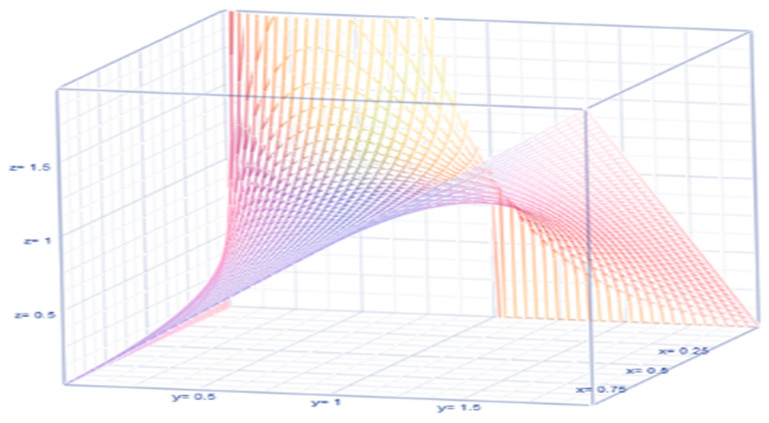
The graph of the probability density function of the COVID-19 deaths in US.

**Figure 2 healthcare-09-01175-f002:**
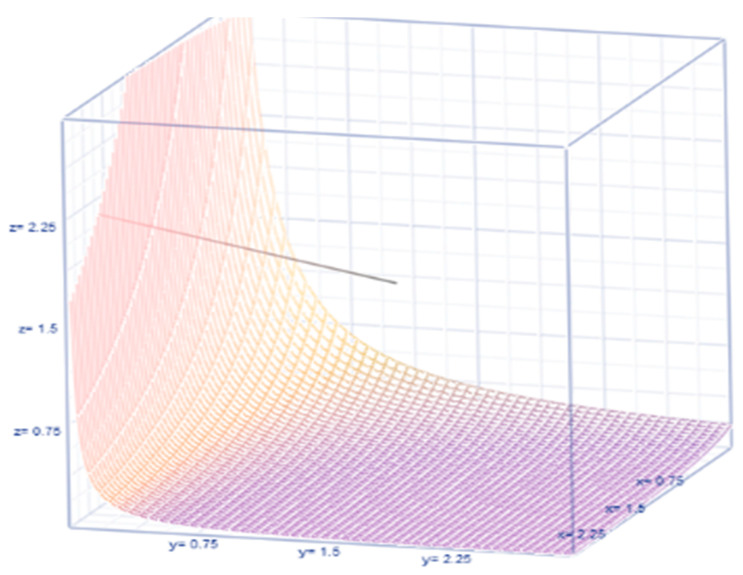
The graph of balancing factor.

**Figure 3 healthcare-09-01175-f003:**
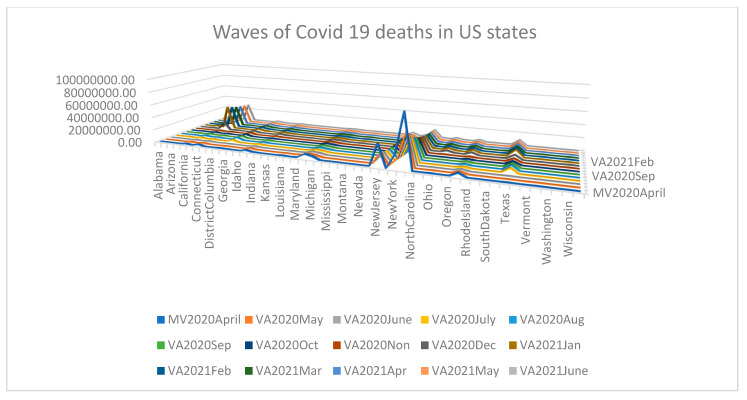
Pandemic waves of COVID-19 deaths in US.

**Figure 4 healthcare-09-01175-f004:**
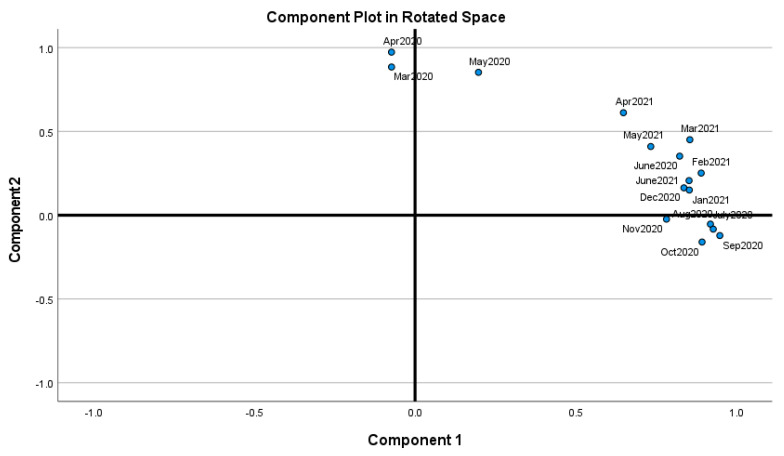
Proximity of the months (June 2020 through June 2021 as Cluster 1 and March, April, May 2020 as Cluster 2) with 81% of total variation explained by two principal components.

**Figure 5 healthcare-09-01175-f005:**
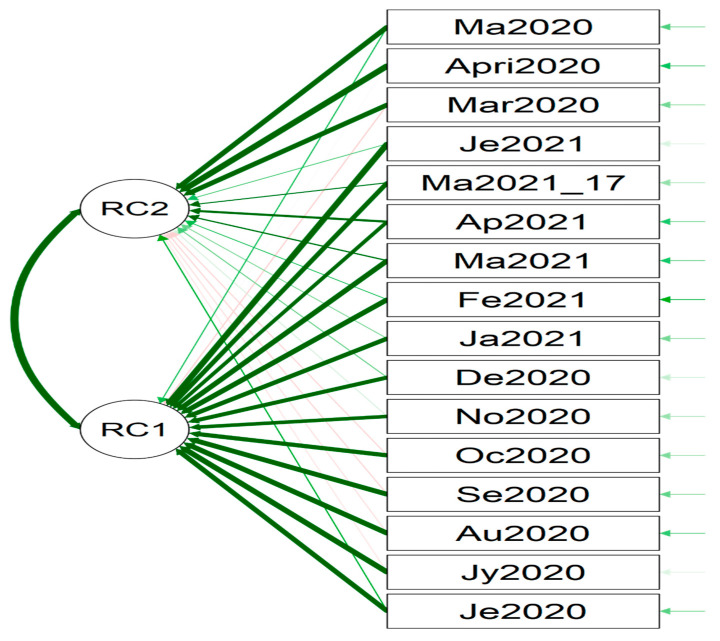
PC1 and PC2 (March, April, May 2020 in PC1, June 2020- June 2021 in PC2). For the readability, note that Fe2021 refers February 2021.

**Figure 6 healthcare-09-01175-f006:**
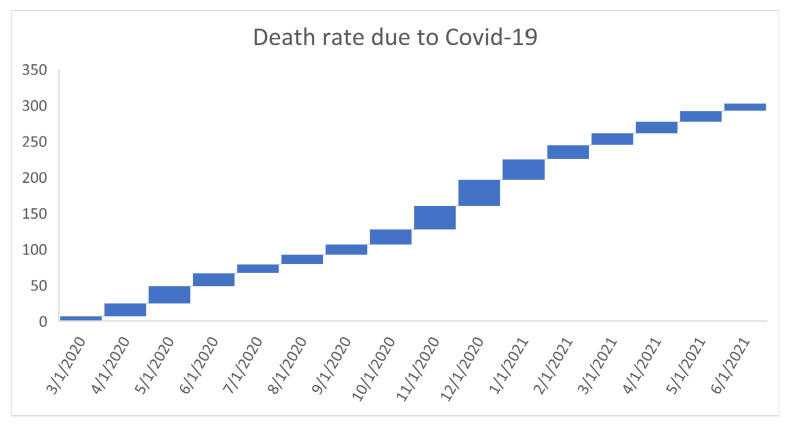
Progressive death rates n US. For readability, note that 4/1/2020 refers 1 April 2020.

**Figure 7 healthcare-09-01175-f007:**
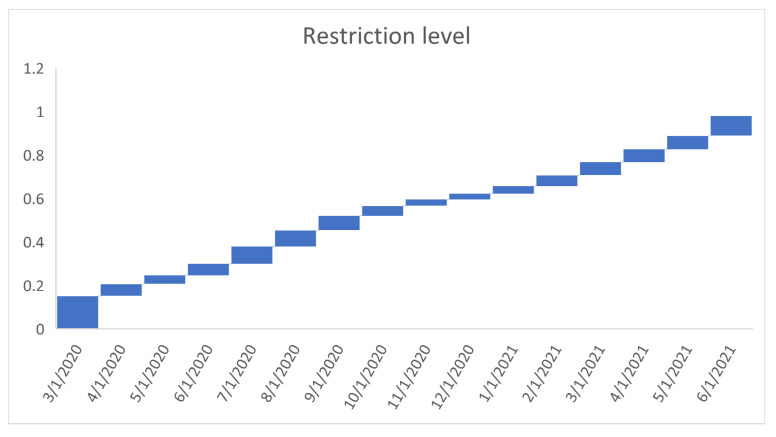
Progressive restriction levels in US. For readability, note that 4/1/2020 refers 1 April 2020.

**Figure 8 healthcare-09-01175-f008:**
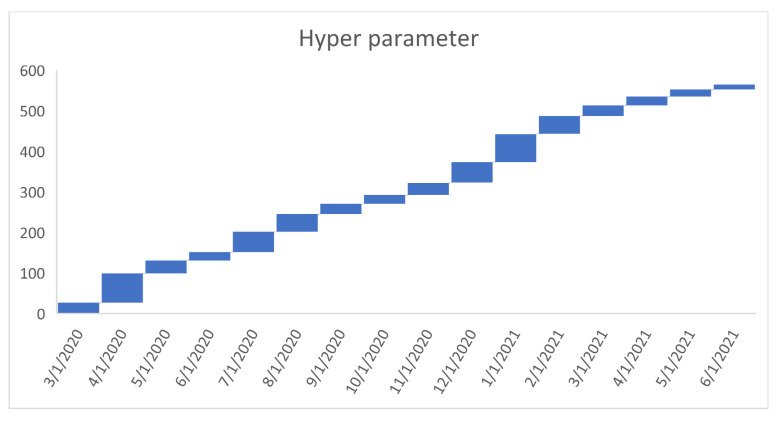
Progressive hyperparameter in US. For readability, note that 4/1/2020 refers 1 April 2020.

**Figure 9 healthcare-09-01175-f009:**
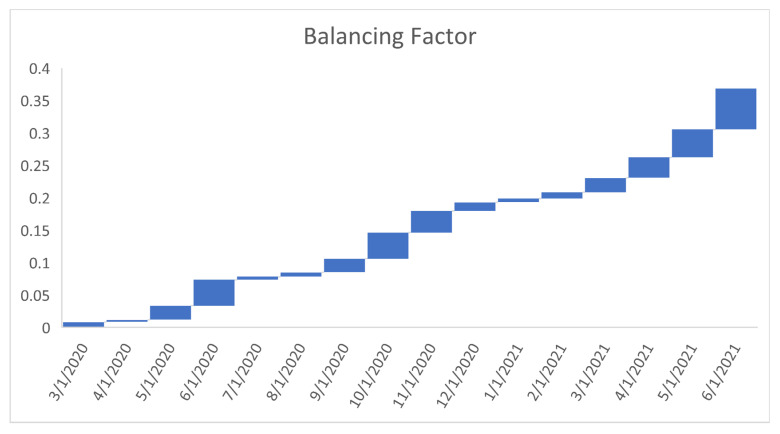
Progressive balancing factor in US. For readability, note that 4/1/2020 refers 1April 2020.

**Figure 10 healthcare-09-01175-f010:**
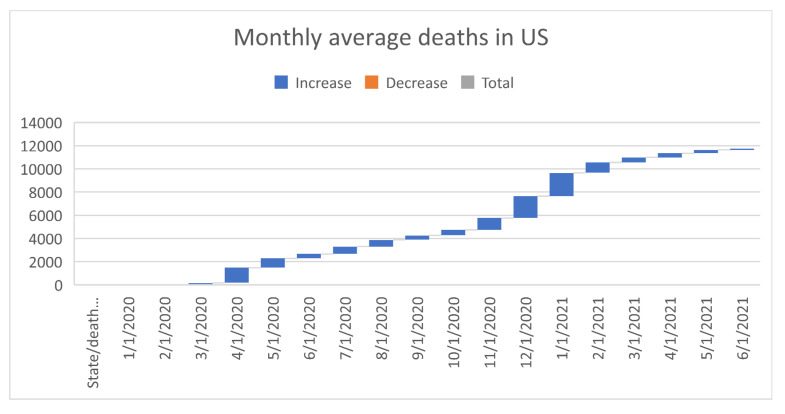
Progressive average number of deaths in US. For readability, note that 4/1/2020 refers 1 April 2020.

**Figure 11 healthcare-09-01175-f011:**
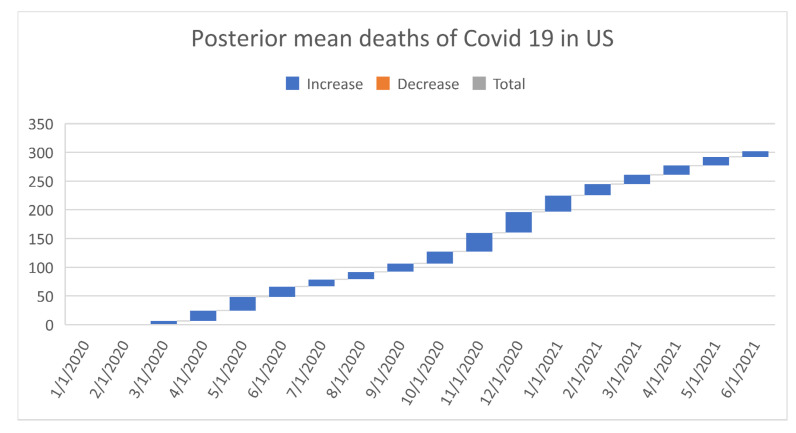
Progressive posteriors mean deaths in US. For readability, note that 4/1/2020 refers 1 April 2020.

**Figure 12 healthcare-09-01175-f012:**
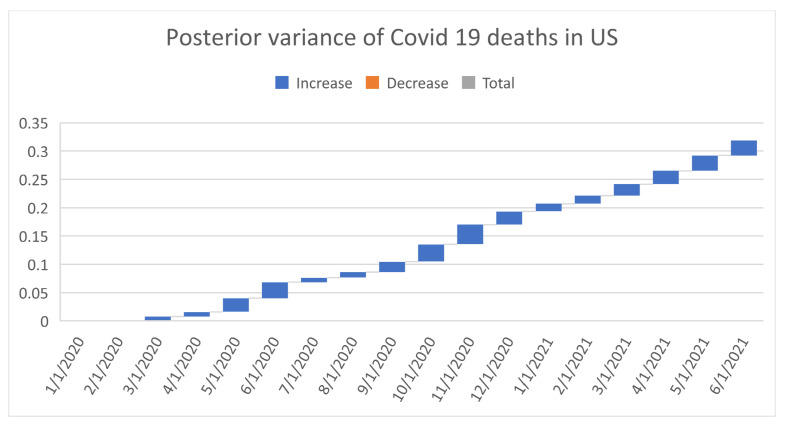
Progressive variance of COVID-19 deaths in US. For readability, note that 4/1/2020 refers 1 April 2020.

**Figure 13 healthcare-09-01175-f013:**
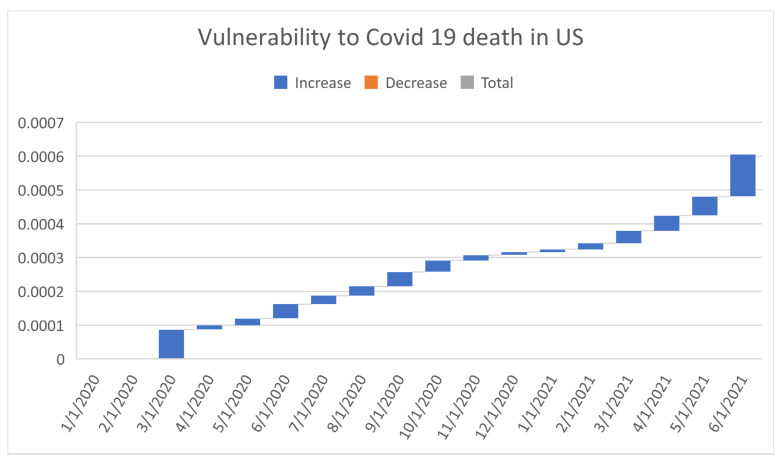
Progressive vulnerability to COVID-19 deaths in US. For readability, note that 4/1/2020 refers 1 April 2020.

**Table 1 healthcare-09-01175-t001:** Estimate of the death rate $\hat{\theta }$, restriction rate $\hat{\gamma }$, hyper parameter $\hat{\phi }$, balance factor, # days n, average # deaths $\bar{y}$, posterior mean, posterior variance, and vulnerability to death. For readability, note that 2/1/2020 refers 1 February 2020.

Scheme	$\hat{\theta }$	$\hat{\gamma }$	$\hat{\phi }$	Balance Factor	N (# Days)	$\bar{y}$	Posterior Mean	Posterior Variance	Vulnerability to Death
1/1/2020	No	data			31				
2/1/2020	No	data			29				
3/1/2020	6.4112	0.15	26.637	0.008	31	177.175	6.443	0.008	8.6852 × 10^−5^
4/1/2020	17.945	0.055	71.942	0.003	30	1308.96	17.98	0.008	1.23654 × 10^−5^
5/1/2020	23.98	0.04	32.262	0.022	31	797.542	24.01	0.024	2.04433 × 10^−5^
6/1/2020	18.02	0.053	20.237	0.04	30	382.745	18.05	0.028	4.22952 × 10^−5^
7/1/2020	12.32	0.08	50.52	0.005	31	634.551	12.35	0.008	2.51761 × 10^−5^
8/1/2020	13.177	0.074	43.448	0.007	31	585.706	13.21	0.01	2.73495 × 10^−5^
9/1/2020	14.36	0.067	25.115	0.021	30	375.039	14.39	0.018	4.28503 × 10^−5^
10/1/2020	20.965	0.046	21.837	0.04	31	478.846	21	0.031	3.39436 × 10^−5^
11/1/2020	32.672	0.03	30.309	0.033	30	1023.13	32.7	0.035	1.60284 × 10^−5^
12/1/2020	36.245	0.027	51.061	0.013	31	1885.27	36.28	0.023	8.71234 × 10^−6^
1/1/2021	28.586	0.034	69.174	0.006	31	2004.06	28.62	0.014	8.1642 × 10^−6^
2/1/2021	19.72	0.05	44.59	0.009	28	899.385	19.75	0.014	1.80436 × 10^−5^
3/1/2021	16.174	0.06	25.94	0.022	31	435.788	16.21	0.02	3.70254 × 10^−5^
4/1/2021	15.92	0.06	21.478	0.032	30	357.922	15.95	0.024	4.50578 × 10^−5^
5/1/2021	15.191	0.062	17.782	0.043	31	285.28	15.22	0.027	5.64437 × 10^−5^
6/1/2021	10.043	0.092	11.648	0.063	30	127.02	10.07	0.027	0.000124612

## Data Availability

The utilized data were downloaded from the webpage https://www.cdc.gov/ (accessed on 1 September 2021).

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
