# Peer review of "A Report Card on Prevention Efforts of COVID-19 Deaths in US"

_healthcare, 2021, doi:10.3390/healthcare9091175_

Round 1

Reviewer 1 Report

The paper seems well concerned and methods of analysis are well explained.

they are clear all the purpose of the calculation in order to reach the result that the Authors pursue.

in this way I think that the conclusion are not sufficient connected with the other partes of the paper.

In practical the assertions the Authors do in the conclusion seems unexpected compared with all the assertion and purposes they described in the general project of the study.

The discussion (really not well marked in the paper) and the  conclusions contains only general assertion not appropriated in a scientific paper and, overall, not so effective from a scientific point of view  when compared with the interesting work and its application exposed in the Method section. 

Author Response

Dear Editor:

Thank you for the opportunity to revise the manuscript. We have now incorporated all the editor’s, referee 1’s and referee 2’s suggestions. The revised manuscript reads much better. The details follow. We appreciate and thank you!

Dr. Ram Shanmugam and co-authors.

Dear Dr. Shanmugam,

Please revise the manuscript according to the referees' comments and upload the revised file before 28 August 2021.

Our response:

The manuscript is revised incorporating all the suggestions given by the referees.

Besides, you need to pay attention to the format of references in the manuscript, please change the superscript into the square brackets (for

example: Chen1 into [1]).

Our response:

As suggested, the superscripts are now changed into the format [] in this revision.

Please use the version of your manuscript found at the above link for your revisions.

Our response:

We did so.

(I) Any revisions to the manuscript should be marked up using the “Track Changes” function if you are using MS Word/LaTeX, such that any changes can be easily viewed by the editors and reviewers.

Our response:

In this revision, we used the “Track Changes” functions as advised.

(II) Please provide a cover letter to explain, point by point, the details of the revisions to the manuscript and your responses to the referees’ comments.

Our response:

Separate letters (one for the editor, one for referee1, one for referee 2) are prepared and submitted with authors’ response for their comments.

(III) If you found it impossible to address certain comments in the review reports, please include an explanation in your rebuttal.

Our response:

The revision is made and submitted.

(IV) The revised version will be sent to the editors and reviewers.

If one of the referees has suggested that your manuscript should undergo extensive English revisions, please address this issue during revision. We propose that you use one of the editing services listed at

https://nam04.safelinks.protection.outlook.com/?url=https%3A%2F%2Fwww.mdpi.com%2Fauthors%2Fenglish&data=04%7C01%7Cshanmugam%40txstate.edu%7Cd2e4deb032f44a6c353308d9620f6f37%7Cb19c134a14c94d4caf65c420f94c8cbb%7C0%7C0%7C637648640865751778%7CUnknown%7CTWFpbGZsb3d8eyJWIjoiMC4wLjAwMDAiLCJQIjoiV2luMzIiLCJBTiI6Ik1haWwiLCJXVCI6Mn0%3D%7C3000&sdata=SHB0fJaEi5ou8LtdGh5AMfzGhs9N75pQEF3dG%2BRlH98%3D&reserved=0 or have your manuscript checked by a native English-speaking colleague.

Our response:

None of the referees suggested extensive English revision. The English and grammar are now thoroughly checked and improved in this revision.

Please note that author names, affiliations and e-mail could not be changed if paper accepted, so please check it carefully when revising your manuscript.

Our response:

The author names, affiliations, e-mail etc. have been checked in this revision and the statements are true.

Do not hesitate to contact us if you have any questions regarding the revision of your manuscript. We look forward to hearing from you soon.

Our response

We appreciate and thank you for the helps. We look forward to having our article is published in the journal Healthcare. It is an honor to publish in this journal.

Reviewer 1

Authors’ response to Reviewer 1 of Manuscript ID: healthcare-1348008 with the title “A report card on prevention efforts of COVID-19 deaths in US”

( ) 
Extensive editing of English language and style required
( ) Moderate English changes required
( ) English language and style are fine/minor spell check required
(x) I don't feel qualified to judge about the English language and style

Yes

Can be improved

Must be improved

Not applicable

Does the introduction provide sufficient background and include all relevant references?

(x)

( )

( )

( )

Is the research design appropriate?

(x)

( )

( )

( )

Are the methods adequately described?

(x)

( )

( )

( )

Are the results clearly presented?

( )

(x)

( )

( )

Are the conclusions supported by the results?

( )

( )

(x)

( )

Comments and Suggestions for Authors

The paper seems well concerned and methods of analysis are well explained.

They are clear all the purpose of the calculation in order to reach the result that the Authors pursue.

In this way I think that the conclusion are not sufficient connected with the other parts of the paper.

In practical the assertions the Authors do in the conclusion seems unexpected compared with all the assertion and purposes they described in the general project of the study.

The discussion (really not well marked in the paper) and the conclusions contains only general assertion not appropriated in a scientific paper and, overall, not so effective from a scientific point of view when compared with the interesting work and its application exposed in the Method section. 

 Our response:

Authors appreciate and thank the referee for suggestions, which we incorporated in this revision. Consequently, the manuscript reads better. Your help is now acknowledged in the manuscript.

  1. In all parts of the manuscript, the English and presentation are improved as you advised.
  2. Please see the addition in line 269 through 284 (in the revised version) to strengthen the conclusion as you advised us. We believe that the addition eliminates the weak discussion (really not well marked in the prior version of the paper) general assertion to effective statements from a scientific point of view, when compared with the interesting work and its application exposed in the Method section. 

The details are:

With an appropriate data, a correct methodology to analyze and extract the evidence to learn about what have gone correctly and what could have been done differently is the essence of the article. As we witnessed in the contents of this article, there had been an increasing death rate due to COVID-19 pandemic and equally compatible stronger impact of medical and social interventions on the death rate. The performances in some states are similar while others differed significantly. However, the deaths in the months March, April, May 2020 clustered together, while all other months deviated into another cluster. This pattern becomes visible due to our data analysis. The death rates and the impact of medical and social interventions had consistently increased together over the months. Such a co-movement complicates our comprehension. There ought to have been a chain adjustment in their relationship and it was captured by the balancing factor. The chain relationship is the underlying reason for considering the aptness of the Bayesian concept and tools as they are used in the article. Interestingly, the balancing factor itself, as expected, consistently increased and it attested to the fact that the health system had been trying to control the pandemics effectively. In spite of the efforts to contain the pandemic, the vulnerability to death due to COVID-19 has been increasing over the months in a volatile manner because of the heterogenous nature of the US states.

We appreciate and thank you for your valuable time and constructive suggestions with encouragements to revise!

Reviewer 2 Report

The goal of this paper is to assess how state-level deaths due to COVID correlate with social distancing restrictions by state, and important and timely issue. Although the model seems to be technically well-estimated, the model is not justified, leaving the reader wondering why simpler regression techniques would not suffice. The components of the model are not explained, so it seems like a ‘black box’. I think with a rewrite would highlight the importance of the estimation work.

-The introduction and conclusion should be better integrated with the model. The introduction not only doesn’t relate to the model that much, but it does not explain why this model and approach differ from the literature.

-A defense of the methods used, probability model justification, in comparison to simpler non-Bayesian approaches is not made. The simpler the model which can be defended, the better. There must be a reason why this model was used. Why was this model used and why is it better?

-What are the independent variables included? Are they state-level SIP restrictions only? Are they lagged? What about mask mandates​? And are urban restrictions also considered, especially in urban states?

-What about local attitudes independent of restrictions? For instance, during the delta surge, roughly 50% are masking and many are not going out, despite the fact that no laws require masking or staying at home. I have observed rural areas with no masking or social distancing. urban

-What about targeted restrictions such as bars as opposed to general SAH?

There is a story to tell with this model.

Author Response

Dear Editor:

Thank you for the opportunity to revise the manuscript. We have now incorporated all the editor’s, referee 1’s and referee 2’s suggestions. The revised manuscript reads much better. The details follow. We appreciate and thank you!

Dr. Ram Shanmugam and co-authors.

Dear Dr. Shanmugam,

Please revise the manuscript according to the referees' comments and upload the revised file before 28 August 2021.

Our response:

The manuscript is revised incorporating all the suggestions given by the referees.

Besides, you need to pay attention to the format of references in the manuscript, please change the superscript into the square brackets (for

example: Chen1 into [1]).

Our response:

As suggested, the superscripts are now changed into the format [] in this revision.

Please use the version of your manuscript found at the above link for your revisions.

Our response:

We did so.

(I) Any revisions to the manuscript should be marked up using the “Track Changes” function if you are using MS Word/LaTeX, such that any changes can be easily viewed by the editors and reviewers.

Our response:

In this revision, we used the “Track Changes” functions as advised.

(II) Please provide a cover letter to explain, point by point, the details of the revisions to the manuscript and your responses to the referees’ comments.

Our response:

Separate letters (one for the editor, one for referee1, one for referee 2) are prepared and submitted with authors’ response for their comments.

(III) If you found it impossible to address certain comments in the review reports, please include an explanation in your rebuttal.

Our response:

The revision is made and submitted.

(IV) The revised version will be sent to the editors and reviewers.

If one of the referees has suggested that your manuscript should undergo extensive English revisions, please address this issue during revision. We propose that you use one of the editing services listed at

https://nam04.safelinks.protection.outlook.com/?url=https%3A%2F%2Fwww.mdpi.com%2Fauthors%2Fenglish&data=04%7C01%7Cshanmugam%40txstate.edu%7Cd2e4deb032f44a6c353308d9620f6f37%7Cb19c134a14c94d4caf65c420f94c8cbb%7C0%7C0%7C637648640865751778%7CUnknown%7CTWFpbGZsb3d8eyJWIjoiMC4wLjAwMDAiLCJQIjoiV2luMzIiLCJBTiI6Ik1haWwiLCJXVCI6Mn0%3D%7C3000&sdata=SHB0fJaEi5ou8LtdGh5AMfzGhs9N75pQEF3dG%2BRlH98%3D&reserved=0 or have your manuscript checked by a native English-speaking colleague.

Our response:

None of the referees suggested extensive English revision. The English and grammar are now thoroughly checked and improved in this revision.

Please note that author names, affiliations and e-mail could not be changed if paper accepted, so please check it carefully when revising your manuscript.

Our response:

The author names, affiliations, e-mail etc. have been checked in this revision and the statements are true.

Do not hesitate to contact us if you have any questions regarding the revision of your manuscript. We look forward to hearing from you soon.

Our response

We appreciate and thank you for the helps. We look forward to having our article is published in the journal Healthcare. It is an honor to publish in this journal.

Reviewer 2

Authors’ response to Reviewer 2 of Manuscript ID: healthcare-1348008 with the title “A report card on prevention efforts of COVID-19 deaths in US”

Authors appreciate and thank the referee for suggestions, which we incorporated in this revision. Consequently, the manuscript reads better. Your help is now acknowledged in the manuscript.

Open Review

English language and style

( ) Extensive editing of English language and style required
(x) Moderate English changes required
( ) English language and style are fine/minor spell check required
( ) I don't feel qualified to judge about the English language and style

Yes

Can be improved

Must be improved

Not applicable

Does the introduction provide sufficient background and include all relevant references?

( )

( )

(x)

( )

Is the research design appropriate?

( )

( )

(x)

( )

Are the methods adequately described?

( )

( )

(x)

( )

Are the results clearly presented?

( )

( )

(x)

( )

Are the conclusions supported by the results?

( )

( )

(x)

( )

Comments and Suggestions for Authors

The goal of this paper is to assess how state-level deaths due to COVID correlate with social distancing restrictions by state, and important and timely issue. Although the model seems to be technically well-estimated, the model is not justified, leaving the reader wondering why simpler regression techniques would not suffice. The components of the model are not explained, so it seems like a ‘black box’. I think with a rewrite would highlight the importance of the estimation work.

Our response:

The goal of this paper is to assess how the number of state-level deaths due to COVID correlate with social distancing restrictions by the state. This is now clearly stated in this revision with statements showing their importance and timely help for future healthcare administrators. The model in the revision is simplified, well-articulated for easy comprehension in this revision. The unsuitability of simpler regression techniques is now stated in the revision. The components of the model are now well explained.

The introduction and conclusion should be better integrated with the model. The introduction not only doesn’t relate to the model that much, but it does not explain why this model and approach differ from the literature.

Our response:

Additional statements are inserted in the introduction and conclusion in this revision. These insertions have well integrated the contents with the model. The introduction in this revision is now well relate to the model and it is now well explaining why our model and approach are better and suitable for the goal.

A defense of the methods used, probability model justification, in comparison to simpler non-Bayesian approaches is not made. The simpler the model which can be defended, the better. There must be a reason why this model was used. Why was this model used and why is it better?

Our response:

Because data on the predictors are not collected, are not available, the regression techniques are not feasible. Hence, we took an approach based on a suitable probability model in comparison to simpler regression and non-Bayesian approaches. Because we noticed the variance is larger than the mean, the regular Poisson model (which requires the equality of mean and variance) is not suitable and a better model called Incidence Rate Restricted Poisson (IRRP) model is suitable alternative and it worked well for the Covid’s death data.

 What are the independent variables included? Are they state-level SIP restrictions only? Are they lagged? What about mask mandates​? And are urban restrictions also considered, especially in urban states?

Our response

The social distancing, lock-down, wearing facemask etc. have an effect on the Covid’s death rate. Such effect is the restriction parameter of the IRRP model. There is only one variable and that is the number of Covid deaths in a US state. There are no independent variables. The effects might have been lagged and we did not study it in this paper but will in another paper. We do not have data on the mask mandates​ and the urban/rural information in the data. Hence, they are not involved in the paper.   

 What about local attitudes independent of restrictions? For instance, during the delta surge, roughly 50% are masking and many are not going out, despite the fact that no laws require masking or staying at home. I have observed rural areas with no masking or social distancing. What about targeted restrictions such as bars as opposed to general SAH?

Our response:

Yes, we agree that there would be local attitudes on the restrictions such as masking, staying at home, or social distancing etc. Because data are not available about them, they are not included in our analysis.  We agree with you on your excellent point about the targeted restrictions such as bars as opposed to general social restrictions. We do not know how many among the dead visited bars. Due to lack of data on visiting bars, it is not included in the model.

There is a story to tell with this model.

Our response:

This is true. The Covid death is different from an ordinary death. The number of ordinary deaths could have a usual Poisson frequency pattern and the text-book Poisson probability model is appropriate. The usual Poisson model requires the equality of mean and variance and it is clearly violated in the Covid’s data. This violation is a symptom exhibited by the data and the root-cause of the symptom is the collective-effect of all preventive measures (like face-masking, social distancing, stay at home etc.) on the Covid’s death rate. Hence, the incidence rate restricted Poisson model is more appropriate and selected to analyze the data.

We appreciate and thank you for your valuable time and constructive suggestions with encouragements to revise!

Round 2

Reviewer 1 Report

I think that Authors answered to the adresses comments and the paper can be published in the present form